# Neprilysin Inhibition in the Prevention of Anthracycline-Induced Cardiotoxicity

**DOI:** 10.3390/cancers15010312

**Published:** 2023-01-03

**Authors:** Aleksandra M. Sobiborowicz-Sadowska, Katarzyna Kamińska, Agnieszka Cudnoch-Jędrzejewska

**Affiliations:** Chair and Department of Experimental and Clinical Physiology, Laboratory of Centre for Preclinical Research, Medical University of Warsaw, 02-091 Warsaw, Poland

**Keywords:** anthracyclines, doxorubicin, cardiotoxicity, neprilysin, LCZ696, sacubitril, valsartan

## Abstract

**Simple Summary:**

Anthracycline-induced cardiotoxicity (AIC) poses a significant clinical challenge in the management of cancer patients. Thus, the development of effective preventive measures for AIC is a heavily studied subject in the field of cardio-oncology. A new class of agents, namely angiotensin receptor/neprilysin inhibitors (ARNi; sacubitril/valsartan), recently incorporated into the management of heart failure with reduced ejection fraction, were found to possess robust cardioprotective effects in preclinical models of a wide range of cardiovascular pathologies. This review discusses the cardioprotective mechanisms of action of sacubitril/valsartan in relation to the pathophysiologic processes involved in the cardiotoxicity of anthracyclines: myocardial remodeling, cardiomyocyte DNA damage, oxidative stress, mitochondrial dysfunction, endoplasmic reticulum stress, inflammatory response, and renin-angiotensin-aldosterone system dysregulation. Additionally, the available data on the effectiveness of sacubitril/valsartan administration in the prevention of AIC were summarized. Several reports on sacubitril/valsartan administration in animal models of AIC published at the time of writing have shown promising results, as ARNi prevented anthracycline-induced myocardial systolic dysfunction and remodeling by alleviating oxidative stress, mitochondrial dysfunction, endoplasmic reticulum stress, and the inflammatory response. Human data remain limited—an ongoing PRADAII trial, aimed to assess the efficacy of sacubitril/valsartan in patients receiving chemotherapy for breast cancer, is expected to be completed in 2025.

**Abstract:**

Anthracycline-induced cardiotoxicity (AIC) poses a clinical challenge in the management of cancer patients. AIC is characterized by myocardial systolic dysfunction and remodeling, caused by cardiomyocyte DNA damage, oxidative stress, mitochondrial dysfunction, or renin-angiotensin-aldosterone system (RAAS) dysregulation. In the past decade, after positive results of a PARADIGM-HF trial, a new class of drugs, namely angiotensin receptor/neprilysin inhibitors (ARNi), was incorporated into the management of patients with heart failure with reduced ejection fraction. As demonstrated in a variety of preclinical studies of cardiovascular diseases, the cardioprotective effects of ARNi administration are associated with decreased oxidative stress levels, the inhibition of myocardial inflammatory response, protection against mitochondrial damage and endothelial dysfunction, and improvement in the RAAS imbalance. However, data on ARNi’s effectiveness in the prevention of AIC remains limited. Several reports of ARNi administration in animal models of AIC have shown promising results, as ARNi prevented ventricular systolic dysfunction and electrocardiographic changes and ameliorated oxidative stress, mitochondrial dysfunction, endoplasmic reticulum stress, and the inflammatory response associated with anthracyclines. There is currently an ongoing PRADAII trial aimed to assess the efficacy of ARNi in patients receiving breast cancer treatment, which is expected to be completed by late 2025.

## 1. Introduction

Anthracyclines are a class of cytotoxic drugs that were introduced into the clinical field in the 1960s [1] and remain widely used in modern cancer chemotherapy [2]. The anthracycline group includes several agents: daunorubicin, epirubicin, idarubicin, and doxorubicin (DOX). They are frequently incorporated into chemotherapeutic regimens for the management of hematologic malignancies such as leukemias and lymphomas; a vast range of solid tumors, i.e., breast, ovarian, bladder, and lung cancers; and soft tissue sarcomas [3]. Because of the scope of malignancies that they are effective against, anthracyclines are also broadly used in the management of childhood cancers [4]. However, anthracycline-based chemotherapy has a number of serious side effects, limiting the lifetime dosage of those drugs [3]. One of the most severe adverse events classically associated with this class of drug is anthracycline-induced cardiotoxicity (AIC). It typically leads to the progressive systolic dysfunction of the left ventricle (LV) and the subsequent development of congestive heart failure (CHF) with its life-long consequences [5].

The clinical challenge posed by the development of AIC in a notable proportion of patients resulted in robust research in an effort to understand the pathophysiology and molecular mechanisms of their cardiotoxicity and to develop effective measures for predicting and preventing it. Accordingly, the cardioprotective effects of coadministration of numerous agents, including the neurohormonal blocking drugs used in the standard management of heart failure (HF), namely angiotensin-converting enzyme inhibitors (ACEi), angiotensin II receptor blockers (ARB), beta-adrenergic receptor antagonists (BB), and aldosterone antagonists, were explored in preclinical and clinical studies [6,7]. In the past decade, a new class of drugs, namely angiotensin receptor/neprilysin inhibitors (ARNi), was incorporated into the management of patients with heart failure with reduced ejection fraction (HFrEF), owing to the practice-changing results of the PARADIGM-HF trial [8]. It was shown that ARNi administration was superior to ACEi–enalapril in reducing the risk of death from cardiovascular causes of such patients [8]. ARNi effectiveness was later explored in the subgroup of patients with HFrEF because of the cardiotoxicity of chemotherapy [9]. The robust cardioprotective effects of ARNi sparked interest in their potential usefulness in the prevention of AIC. The aim of this review is to explore the current understanding of the mechanisms behind ARNi cardioprotection, to determine their relation to the pathophysiologic processes involved in the cardiotoxicity of anthracyclines, and to summarize the available data on their effectiveness in the prevention of AIC.

## 2. AIC Characteristics

Until recently, the definitions and classifications of cancer therapy–related cardiotoxicity varied between clinical societies, especially in terms of the threshold of clinically important LV dysfunction, impairing the ability to directly compare cardiotoxicity rates between studies [10]. In an effort to address this issue, the International Cardio-Oncology Society in the year 2022 published a consensus statement uniformly defining cardiovascular toxicities of cancer therapies [11]. The term cancer therapy–related cardiac dysfunction (CTRCD) encompasses symptomatic HF and asymptomatic cardiac dysfunction related to chemotherapy [11]. The latter is further defined on the basis of left ventricle ejection fraction (LVEF) changes and graded accordingly: a decrease in LVEF to <40% is graded as severe and 40–49% as moderate, whereas mild asymptomatic CTRCD is defined as preserved LVEF (≥50%) with >15% reduction in global longitudinal strain relative to baseline or a new rise in cardiac troponins and/or natriuretic peptides (NPs) [11]. According to the 2022 European Society of Cardiology Guidelines on cardio-oncology, the preferred imaging modality to diagnose and monitor CTRCD is three-dimensional transthoracic echocardiography (TTE) and, when not feasible, two-dimensional TTE or cardiac magnetic resonance (CMR) are of use [12]. According to those guidelines, AIC could be defined as CTRCD or other forms of cardiovascular toxicities (myocarditis, cardiac arrhythmias) related to chemotherapy with anthracyclines, which can present clinically or be detected during surveillance imaging [12].

AIC can present early after the administration of anthracyclines, but what is notable about this adverse event is that symptoms may appear decades after treatment cessation [13]. Classically, AIC is divided into two categories depending on the time of the onset of the first symptoms. There is an acute form of AIC, which occurs within days of anthracycline administration and is usually considered to be reversible [5]. It is associated with the development of myopericarditis or involvement of the cardiac conducting system, leading to the occurrence of arrhythmias; thus, when symptomatic, acute AIC typically presents with chest pain and palpitations [5,14]. The other, chronic form of AIC occurs later, often after the treatment cessation, as CTRCD—presenting as HFrEF or in the asymptomatic form detected on cardiac imaging [5]. Chronic AIC usually becomes clinically apparent within the first year of the treatment completion (early-onset chronic cardiotoxicity), but it may also occur many years after chemotherapy has been completed (late-onset chronic cardiotoxicity) [5]. However, there is currently a tendency to consider those different forms of acute and chronic AIC as manifestations of a continuous cardiotoxic process at different time points rather than as distinct phenomena [5,15].

Our understanding of the scope of the cardiac toxicity of anthracyclines has substantially evolved since their introduction to the clinical practice. In an early retrospective study by Von Hoff et al. (1979) on the epidemiology of AIC in adults, the estimated incidence of AIC, defined as the presence of clinical manifestations of CHF, was 2.2% [16]. A different retrospective study by Swain et al. (2003), including DOX-treated patients from three clinical trials, in which LVEF was measured by equilibrium radionuclide angiography, has shown that the incidence of CHF or asymptomatic decline in LV function after anthracycline therapy is probably higher, estimated at around 5.1% [17]. Nowadays, the introduction of sophisticated cardiac imaging technologies, such as CMR imaging, revealed that the incidence of AIC is even higher than when measured with TTE alone [18].

The issue of chronic AIC is especially challenging in childhood cancer survivors, and DOX treatment is applied in a wide variety of pediatric cancers [4]. The tremendous progress in the management of pediatric cancers increased the survival rates, leading to a growing population of patients with a history of childhood DOX exposure, bearing an elevated risk of cardiovascular complications later in life [19]. In a large retrospective analysis of 14,358 cases of childhood cancer survivors with a history of DOX exposure, they were shown to have a significantly higher risk of developing CHF, myocardial infarction, pericardial disease, or valvular abnormalities [20]. In this study, the incidence of CHF 30 years after cancer diagnosis was 4.1%, which was six times higher than in their unexposed siblings [20]. Among childhood cancer survivors, chronic AIC usually presents clinically in their 20s, 30s, and 40s [20], with the survival rate 10 years after the diagnosis remaining below 50% [19].

The most significant risk factor associated with the development of chronic AIC is the cumulative dose of anthracyclines received [2]. In the aforementioned study by Von Hoff et al. (1979), the percentage of patients who developed CHF at a cumulative DOX dose of 400 mg/m^2^ was 3%, which increased to 7% at 550 mg/m^2^ and further to 18% at 700 mg/m^2^ [16]. Similarly, the aforementioned study by Swain et al. (2003) upheld the strong positive correlation between the cumulative dose of DOX and the risk of CHF development—5% risk at a cumulative dose of 400 mg/m^2^, rising to 16% at a dose of 500 mg/m^2^, 26% at a dose of 550 mg/m^2^, and up to 48% at a dose of 700 mg/m^2^ [17]. This observation was confirmed by several subsequent studies, showing a sharp increase in the risk of both myocardial systolic dysfunction visualized in echocardiography and the development of clinical signs and symptoms of CHF with increasing total DOX dose [21,22]. Other risk factors of chronic AIC include pre-existing heart disease, older age, female sex, arterial hypertension, dyslipidemias, exposure to radiation, and the coadministration of other cardiotoxic agents, i.e., trastuzumab [13].

## 3. Molecular Mechanisms of AIC

On the cellular level, AIC is associated with degenerative changes in cardiomyocytes, leading to their vacuolization, with partial or total loss of myofibrils and the concomitant distention of T-tubules and sarcoplasmic reticulum, as well as the disorganization of their nuclei [23,24,25]. Another classic finding is widespread: patchy myocardial interstitial fibrosis accompanied by fibroblast proliferation and infiltration of histiocytes [23]. In a study by Cove-Smith et al. (2014) on a Hannover Wistar rat model of chronic AIC, where animals received a total of eight doses of DOX (1.25 mg/kg weekly), it has been shown that subcellular changes and significant mitochondrial degeneration are present even after the first dose of DOX, followed by neutrophilic and lymphoplasmacytic infiltration of the myocardium [24]. With subsequent doses, cardiomyocyte degeneration with hypertrophy and extensive vacuolation occurs, coinciding with progressive functional decline, followed by extensive replacement fibrosis after several weeks of doxorubicin treatment [24]. Molecular mechanisms involved in the development of AIC are summarized in Figure 1.

The cytotoxic effects of anthracyclines are related to their ability to damage cellular deoxyribonucleic acid (DNA) in several different mechanisms [26]. They are able to directly intercalate into DNA, interfering with replication and transcription processes [26]. Additionally, anthracyclines target topoisomerase II (Top2), an intranuclear enzyme that manages DNA tangles and supercoils by catalyzing controlled cuts of both DNA strands, preferentially expressed in cells undergoing rapid divisions [27]. Interfering with Top2 activity results in bulky DNA adducts, crosslinks, and double-strand breaks, which trigger cellular apoptosis [27]. However, one of the Top2 isoforms, Top2β, is expressed in quiescent tissues such as cardiomyocytes, allowing for undesirable anthracycline-induced DNA damage to harm those cells [28]. Accordingly, Top2β knockout mice were shown to be less susceptible to AIC [29].

Anthracyclines’ anticancer activity is also associated with their ability to induce cytotoxic levels of oxidative stress [26], achieved by the mitochondrial formation of large amounts of reactive oxygen species and free radicals unmatched by antioxidant defense mechanisms [30]. Anthracyclines interact with mitochondrial nicotinamide adenine dinucleotide phosphate (NADPH) dehydrogenase in complex I of the mitochondrial electron transport chain, subsequently leading to the formation of hydrogen peroxide and hydroxyl radicals, resulting in substantial damage to the inner mitochondrial membrane [26]. Additionally, doxorubicin induces free-radical formation by interacting with nitric oxide synthase (NOS) and iron ions, reduces the endogenous antioxidant activity of catalase and glutathione (GSH), and disrupts the inner mitochondrial membrane by interacting with cardiolipin, all of which exacerbate the oxidative stress levels [30]. Oxidative stress is a well-recognized factor involved in the pathogenesis of different cardiac pathologies [31]. It leads to DNA damage, protein structural modifications, and intracellular lipid peroxidation, resulting in cardiomyocyte dysfunction and death through apoptosis [31]. Reactive oxygen species can also cause the dysregulation of metalloproteinases (MMPs) activity, leading to the activation of profibrotic pathways in the myocardium [32,33].

Additionally, AIC has been associated with mitochondrial damage beyond oxidative stress. Doxorubicin impairs mitochondrial function by promoting mitochondrial DNA disruption [34] and improper mitochondrial fission, leading to a decreased pool of healthy mitochondria in cardiomyocytes [35]. Other mechanisms associated with AIC include endoplasmic reticulum stress [36], the impairment of cardiac calcium homeostasis [37], the activation of ubiquitin-proteasome system-mediated proteolysis [38], autophagy [39], and accelerated cardiomyocyte senescence [40]. Additionally, the renin-angiotensin-aldosterone system (RAAS) seems to be heavily involved in AIC pathophysiology, as anthracycline treatment dysregulates RAAS gene expression, stimulates deleterious angiotensin II (Ang II) signaling through angiotensin II receptor type 1 (ATR1), and downregulates the cardioprotective pathways related to angiotensin II receptor type 2 (ATR2) [6]. Additionally, doxorubicin administration was found to decrease the levels of circulating angiotensin 1-7 (Ang 1-7) and to decrease the myocardial expression of the Mas receptor (MasR), which constitutes the ACE2/Ang 1-7/MasR axis regarded as the cardioprotective arm of the RAAS [6].

Furthermore, apart from the direct damage to cardiomyocytes, AIC is also related to the dysfunction of other cell types present in the myocardium. It was shown that doxorubicin administration causes the depletion of the cardiac progenitor cell pool [41], the disruption of the communication between cardiomyocytes and endothelial cells by alteration of the cytokine profile they release (i.e., endothelin-1 (ET-1), nitric oxide (NO), prostaglandin I2 or neuregulin-1) [42], and the proliferation and enhanced cell survival of fibroblasts [43].

In recent studies, AIC has also been associated with the overactivation of the inflammatory response within the myocardium related to the nuclear-factor kappa-light-chain-enhancer of activated B cells (NF-κB) signaling, with an increase in M1 macrophages infiltrating the cardiac muscle [44,45]. Additionally, toll-like receptor 2 (TLR-2) and 4 (TLR-4), members of a family of pattern recognition receptors taking part in innate immunity, with the ability to induce NF-κB signaling by forming complexes with myeloid differentiation primary response 88 (MyD88), have been recently shown to be upregulated in the myocardium of experimental animals [46] and serum of patients receiving chemotherapy regimens including DOX [47]. Their involvement in AIC pathogenesis was further supported by a recent study on mice models of AIC, where it was shown that TLR-2 knockout prevented the DOX-induced cardiac inflammatory response [48]. The TLR-4/MyD88/NF-κB signaling has been shown to induce the production of several proinflammatory factors and induce the inflammatory response in the myocardial tissues in several studies [49,50]. One example is the TLR-4/MyD88/NF-κB-induced formation of NLR family pyrin domain containing 3 (NLRP3) inflammasome, which has been previously associated with the anthracycline-induced myocardial inflammatory response [51].

## 4. Prevention of AIC

As AIC is a major issue hampering the usefulness of anthracyclines in oncological treatment, efforts have been made to develop methods of AIC prevention. Those include both primary measures, which are applied before or during cancer treatment and are supposed to prevent the development of AIC, and secondary measures, which aim to prevent the progression of HF in cancer survivors who were found to have CTRCD [52]. As for the secondary measures, the European Society of Cardiology recommends LVEF assessment before and within 12 months after the administration of anthracyclines, with additional assessments throughout the treatment for high-risk patients [12]. Managing patients with CTRCD should be based on the clinical guidelines on heart failure [12].

The main interest of this review involves primary preventive measures, which are aimed at the pathophysiological processes underlying the development of AIC. The mainstay of AIC prevention is the management of cardiovascular risk factors such as obesity, cigarette smoking, arterial hypertension, physical activity, and alcohol consumption, according to the European Society of Cardiology Guidelines on cardiovascular disease (CVD) prevention [53], before, throughout, and after chemotherapy [12]. Strategies aimed at decreasing the risk of AIC also include the use of the liposomal formulation of doxorubicin and careful consideration of the administration of dexrazoxane, ACEi, ARB, beta-adrenergic receptor antagonists, and statins in high-risk patients [12].

At the time of writing, the only agent approved by the US Food and Drug Agency and the European Medicines Agency for the primary prevention of AIC is dexrazoxane, in both adult and pediatric populations. The coadministration of dexrazoxane with anthracyclines has several potential mechanisms associated with the cardioprotective effects observed in both preclinical and clinical settings [54]. Dexrazoxane binds to Top2β, changing its configuration, depriving circulating anthracyclines of their molecular target, and as a result preventing DNA damage in cardiomyocytes [55]. Additionally, dexrazoxane is an iron chelator, preventing the interaction between anthracyclines and iron ions and thus reducing oxidative stress levels [56]. Several large-scale meta-analyses have confirmed the statistically significant benefit of dexrazoxane use during anthracycline therapy when compared with no preventive measures, showing a relative risk of HF occurrence between 0.21 and 0.31 [57,58,59]. However, one of the randomized clinical trials of dexrazoxane vs. placebo in breast cancer patients receiving anthracycline-based chemotherapy reported decreased tumor objective response rates in the dexrazoxane-receiving group (46.8% for dexrazoxane and 60.5% for placebo, 95% confidence interval: −25% to −2%; *p* = 0.019), without any changes to overall survival and progression-free survival [60]. Though this finding was not supported by several meta-analyses and expert statements, underlining no change in the antitumor efficacy of anthracyclines when coadministered with dexrazoxane, it was associated with decreased enthusiasm for the use of dexrazoxane in adult cancer patients [61,62,63].

The efficacy of neurohormonal blocking drugs, such as ACEi, ARB, aldosterone antagonists, and BB in the prevention of AIC, is currently being excessively studied. Results from numerous preclinical studies in animal models have shown that those agents are potentially effective and can prevent or reduce the severity of AIC [6,54]. This was later upheld in some clinical trials, where enalapril, perindopril, ramipril, valsartan, telmisartan, candesartan, spironolactone, carvedilol, and nebivolol have been shown to exhibit potential cardioprotective effects, mainly preventing the hemodynamic abnormalities induced by the anthracyclines, visualized by TTE [6,54]. The PRADA trial assessed the efficacy of the administration of an ARB candesartan or of BB metoprolol in breast cancer patients receiving adjuvant anticancer therapy [64]. It was shown that treatment with candesartan decreased the overall decline in LVEF measured by CMR when compared with placebo, which was not observed in the metoprolol-receiving group. However, the OVERCOME trial, assessing cotreatment with ACEi enalapril and BB carvedilol in patients receiving anthracycline-based chemotherapy for hematologic malignancies, reported mixed results, as a decline in LVED was not prevented in patients undergoing autologous hematopoietic stem cell transplantation [65]. Thus, it remains somewhat unclear whether neurohormonal blocking agents provide definite cardioprotection against AIC, and more clinical trials are required.

Other agents, including metformin, statins, and phytochemicals such as resveratrol, allicin, lycopene, curcumin, and polyphenols, showed cardioprotective and anti-inflammatory effects in preclinical AIC models; however, the data on their effectiveness in human trials are limited [52,66,67]. Additionally, there has recently been a growing interest in the potential efficacy of sodium-glucose cotransporter 2 (SGLT-2) inhibitors in AIC prevention, thanks to their antioxidant and anti-inflammatory properties. Quagliariello et al. (2021) demonstrated that SGLT-2 inhibitor empagliflozin alleviated the DOX-induced decrease in LVEF and radial and longitudinal strains in nondiabetic mice [68]. This was associated with the decreased activity of the aforementioned MyD88/NF-κB/NLRP3 proinflammatory pathway and a decrease in myocardial ferroptosis and apoptosis. 

Overall, the quality of data on the primary prevention of AIC remains poor, as most data are derived from small sample studies with short follow-ups [54]. Furthermore, the optimal strategy to prevent AIC remains unknown, as different pharmacological preventive measures were not compared for their effectiveness, rate of adverse effects, and influence on the quality of life of cancer survivors [54]. Finally, as the issue of AIC still poses a clinical challenge in a growing population of patients with a history of anthracycline exposure, the continued development of more-effective and safer preventive measures is required.

## 5. Neprilysin Inhibition in HF

Neprilysin (NEP, neutral endopeptidase, CD10) is a zinc-dependent metalloproteinase involved in the regulation of the activity of many endocrine systems [69]. It is expressed in a membrane-bound form in a wide variety of tissues, including the kidneys, brain, heart, lungs, gastrointestinal system, adrenal glands, reproductive system, and placenta [70]. NEP’s active site is located in the extracellular space, providing a proteolytic cleavage of various substrates in the cellular microenvironment [71]. Additionally, NEP can be excreted in a soluble catalytically active form and is present in the plasma, urine, and synovial fluid [71]. Clinical data suggest that the heart, especially LV in the setting of HF, is a major source of soluble circulating NEP [72], and its levels are positively correlated with an increased risk of hospitalization or death from cardiovascular causes in HF patients [73].

NEP has a variety of known substrates, including NPs, bradykinin, substance P, adrenomedullin, Ang I, Ang II, ET-1, b-amyloid peptide, and somatostatin [55]. This list is constantly being updated as new polypeptides undergoing cleavage by NEP are identified. Recently, a new substrate of NEP, apelin, was described, adding the apelinergic system to the robust network of endocrine systems regulated by NEP [74].

The interest in the potential applicability of neprilysin inhibition in the management of cardiovascular diseases arose from its ability to regulate the natriuretic peptide system (NPS). There are six NPs currently described: atrial NP (ANP), brain NP (BNP), C-type NP (CNP), D-type NP, ventricular NP, and urodilatin [75]. NPS acts as the main counterpart to the leading neurohormonal pathways involved in the pathogenesis of HF–RAAS and the autonomic nervous system. NPs exert their effect by binding to transmembrane NP receptors (NPR), which in turn activate guanylyl cyclase (GC) receptors and lead to the synthesis of secondary messenger cyclic guanosine monophosphate (cGMP) that activates protein kinase G (PKG) [75]. NPs exhibit a wide range of physiological, cardioprotective effects—they cause natriuresis and vasodilation, as well as possess antiproliferative, antifibrotic, and antihypertrophic effects on the cardiomyocytes [76,77]. ANP exerts its effects mainly by regulating the activity of the renal system—it increases the glomerular filtration rate, decreases sodium and water reabsorption, and reduces renin secretion [78]. ANP is also an antagonist to the mineralocorticoid receptor, and together with BNP, they possess a direct renin-inhibiting effect [79]. In various in vitro and animal studies, both ANP and BNP were shown to inhibit cardiomyocyte hypertrophy and fibroblast proliferation induced by RAAS through their interaction with NPR type A (NPRA) [80]. This signaling pathway counteracts the effects of Ang II and attenuates transforming growth factor beta 1 (TGFβ1) in an extracellular signal-regulated kinase (ERK)–dependent manner, as well as inhibits the expression of hypertrophy-related regulators, i.e., transcription factor GATA4 [80].

NPS cardioprotective effects led to the reasoning that the inhibition of NEP, in turn causing increased activity in NPs, would provide substantial benefits in patients with HF. However, NEP’s influence on the cardiovascular system is twofold, as apart from degrading cardioprotective agents such as NPs, it additionally modulates RAAS activation. It cleaves Ang II to inactive peptides, inhibiting the deleterious Ang II/ATR1 signaling. Simultaneously, NEP is involved in the cleavage of Ang I to Ang 1-7, promoting the activation of the Ang 1-7/MasR cardioprotective axis [81]. Thus, the administration of NEP inhibitors (NEPi) alone leads to unopposed Ang II/ATR1 signaling activation, which is considered to be the main reason why clinical trials assessing monotherapy with NEPi have failed to prove significant clinical benefits [82,83]. However, this inspired an effort to combine NEPi with ACEi to counter the overactivation of the RAAS. While this drug combination was successful in providing a survival benefit to HF patients, it was also associated with an increased risk of angioedema owing to a significant buildup of bradykinin, no longer cleaved by neither ACE nor NEP [84]. Finally, a combination drug of NEPi-sacubitril and ARB-valsartan was introduced (angiotensin receptor/neprilysin inhibitor, ARNi, sacubitril/valsartan, LCZ696), offering the inhibition of both NEP and RAAS, without causing significant side effects. ARNi’s combined effects are represented in Figure 2.

The aforementioned PARADIGM-HF clinical trial assessed the effects of treatment with ARNi in comparison to enalapril in patients with class II, III, or IV HR with EF ≤ 40% [8]. The trial was stopped early because of the clear superiority of ARNi over enalapril in decreasing the risk of hospitalization from HF exacerbation or death from cardiovascular causes [8]. Since then, the administration of ARNI has been incorporated as the mainstay of HFrEF management. As for the patients with HF with preserved EF (HFpEF), a large multicenter clinical trial PARAGON-HF comparing ARNi to ARB monotherapy failed to demonstrate differences in the risk of cardiovascular death or total hospitalizations, and thus, the role of ARNi in the treatment of HFpEF remains uncertain [85].

Ever since ARNi emerged into clinical practice, there has been an enormous interest in the precise mechanisms related to its cardioprotective effects. As NEP is involved in the regulation of various endocrine systems, its inhibition was suspected to provide cardioprotection beyond its influence on the NPS. The NEP gene is highly conserved in mammalian species, making neprilysin a convenient candidate to study in preclinical animal models [86]. Thus, this review summarizes the available data from preclinical studies of ARNI in various CVDs, which have expanded the current understanding of their cardioprotective mechanisms of action in different pathophysiological settings.

## 6. ARNi in Preclinical Models of Myocardial Infarction

The administration of ARNi has been excessively studied in preclinical models of myocardial infarction (MI). There is an overlap in the pathophysiologic processes underlying cellular damage during the acute phase of MI and reperfusion injury, and acute myocardial toxicity during the administration of anthracyclines, as both settings include oxidative stress, an inflammatory response, mitochondrial damage, or the dysregulation of ion-channel proteins resulting in susceptibility to arrhythmias [87]. Additionally, myocardial remodeling and hypertrophy associated with post-MI HF are also classically observed in chronic AIC [88]. Thus, the animal studies on ARNi’s effect on the severity of MI and the subsequent development of systolic heart failure are valuable sources of information on the potential effectiveness of ARNi in alleviating pathophysiologic processes related to AIC. Additionally, as many of those studies include experimental groups receiving monotherapy with either ARB or ACEi, they allow for conclusions regarding the additive benefit of concomitant administration of NEPi. The study designs and the most important findings of those studies are summarized in Table 1 [89,90,91,92,93,94,95,96,97,98,99,100,101]. Additionally, details of the experimental animals, protocols of oral drug gavage, and drug dosages are provided in Appendix A.

As shown, in animal models of MI, either ARNi treatment was administered during the acute ischemia and reperfusion and continued throughout several weeks after MI, or it was initiated after the post-MI systolic dysfunction of LV had already developed. The former protocol allows for conclusions on ARNi’s influence on MI severity and post-MI HF development, and the latter provides answers regarding its effect on myocardial systolic dysfunction and remodeling associated with chronic post-MI HF. Treatment with ARNi during the acute phase of MI and that continued throughout the reperfusion has been shown in several studies to offer robust infarct-sparing benefits.

The majority of the reviewed studies that assessed the efficacy of ARNi administration in the acute phase of MI reported protection against post-MI myocardial systolic dysfunction (decline in LVEF, LV dilation), which was frequently not observed in the groups receiving ARB or ACEi alone [89,90,92,94,95,98]. Raj et al. (2021) demonstrated that those beneficial effects were associated with decreased levels of oxidative stress as measured by malondialdehyde (MDA) levels [95]. Additionally, treatment with ARNi decreased the expression of profibrotic and proinflammatory factors such as collagen and tumor necrosis factor α (TNFα). Ishii et al. (2017) found that, in contrast to the enalapril-receiving group, ARNi administration alleviated post-MI decline in LV systolic function and prevented mice morbidity thanks to ventricular muscle rupture as an acute complication of MI [91]. Additionally, on the third day after MI, the mRNA expression of interleukin-1β (IL1β), interleukin-6 (IL6), and matrix MMP-9, as well as MMP-9 activity, were significantly lower in the infarcted myocardium in the ARNi-receiving animals. Taken together, these data suggest that ARNi treatment could suppress an excessive inflammatory response and extracellular matrix degradation due to metalloproteinase activity, which provided protection against post-MI cardiac rupture. In a similar setting, Liu et al. (2021) compared the effects of ARNi administration with those of another ACEi, namely benazepril or combination therapy with both drugs, even though concomitant treatment with ARB and ACEi is controversial [93]. It was shown that ARNi alone or in combination therapy was effective in preserving LV systolic function and ameliorating cardiac hypertrophy and fibrosis 4 weeks after MI. Interestingly, animals receiving either ARNi or combination therapy showed significantly and similarly decreased TGFβ1 levels in cardiac tissue, which was not observed in benazepril monotherapy. This finding suggests that the described beneficial antifibrotic effect may be mostly or solely related to NEP inhibition. This is further supported by Suematsu et al. (2016), who have shown that ARNi treatment resulted in a significantly lower myocardial fibrosis associated with the decreased expression of TGFβ mRNA in cardiomyocytes, which was not seen in mice receiving valsartan monotherapy [97].

The other animal studies of MI available in the literature review adopted a protocol where ARNi and other treatments were administered after several days or weeks after MI induction or when post-MI LV systolic dysfunction was documented by TTE. Such a study design allows for conclusions on ARNi’s effectiveness in alleviating post-MI systolic dysfunction and ventricular remodeling when systolic dysfunction has already developed. Von Lueder et al. (2015) provided an initial report describing the benefit of ARNi administration in this setting [101]. ARNi administration resulted in a significantly improved LV systolic function and decreased ventricular hypertrophy, as well as ameliorated myocardial fibrosis in peri-infarct and remote myocardium. Similarly, in the aforementioned study by Torrado et al. (2018), ARNi treatment in post-MI HFrEF resulted in a significant improvement in LVEF in comparison with the placebo- and valsartan-receiving animals 4 weeks after MI and in comparison with the placebo group 10 weeks after MI [98]. In a different study, by Pfau et al. (2019), ARNi has been shown to induce angiogenesis within the infarct zone [94]. Similar results on the beneficial effect of ARNi treatment on the severity of post-MI systolic dysfunction and myocardial fibrosis were reported by Shen et al. (2021) [96]. Additionally, they reported on the anti-inflammatory and antioxidative effects of ARNi treatment: in the experimental animals, there were decreased levels of serum IL1βa and interleukin-18 (IL18), decreased activity of the NLRP3 inflammasome, and a decreased level of myocardial ROS accumulation. A different study, by Kompa et al. (2018), involved an assessment of its action on metalloproteinases activity [92]. In contrast to the previously described study by Ischii et al. (2017) [91], no significant effects following active treatment with ARNi on MMP-9 activity were reported [92]. On the contrary, ARNi administration was shown to decrease the activity and gene expression of the tissue inhibitor of metalloproteinases 2 (TIMP2), an inhibitor of MMPs that prevents matrix degradation, suggesting a potential mechanism for the enhanced resolution of fibrosis upon ARNi treatment. The reason behind those seemingly contrary results may be associated with the dynamic changes in the pathophysiologic processes involved in MI progression, as those studies included molecular analyses of myocardial tissue either shortly after acute ischemia (Ischii et al. [91]) or after several weeks post-MI (Kompa et al. [92]). A different study, by Trivedi et al. (2018), did not show any significant changes in the severity of myocardial fibrosis upon treatment with ARNi [99]. However, ARNi was superior to valsartan in preventing the diastolic dysfunction of the LV, which was linked to the potent protection of vascular endothelial cells, leading to improved vascular compliance and increased NO bioavailability.

Two of the studies on ARNi administration in post-MI HF focused on its effects on the occurrence of arrhythmias, compared with either enalapril or valsartan monotherapy [89,90]. Both studies have shown that treatment with ARNi was associated with decreased ventricular arrhythmias inducibility. This was linked to the alleviation of ion-channel remodeling that plays an important role in the post-MI electrophysiological changes, as ARNi-treated animals showed upregulation in the myocardial expression of several K+ channel proteins (potassium voltage-gated channel subfamily E member 1, KCNE1; potassium voltage-gated channel subfamily E member 2, KCNE2; and ether-à-go-go–related gene channels, ERG), which was not observed in the other experimental groups [89].

Taken together, studies of ARNi administration in animal models of MI have all shown that there is a significant beneficial effect of such treatment when applied either during the acute phase of MI, where it can attenuate the immediate decrease in LVEF and prevent the development of HFrEF, or later on, in order to improve ventricular systolic dysfunction and alleviate myocardial fibrosis and hypertrophy associated with post-MI HF. In all the reviewed studies, ARNi administration was associated with a significant increase in LVEF values when compared with placebo or, in many cases, also with ARB or ACEi monotherapy, as presented in Table 1. Additionally, ARNi treatment after MI was consistently reported to prevent myocardial fibrosis and cardiomyocyte hypertrophy, and those effects were usually more pronounced than in animals receiving valsartan monotherapy. The functional and molecular benefits of treatment with ARNi were demonstrated to be associated with decreased oxidative stress levels, halted excessive inflammatory response in the myocardium, and the modulation of metalloproteinases activity, all of which are important components of the AIC pathophysiology.

## 7. ARNi in Preclinical Models of Other CVDs

Apart from the previously described studies on MI models, the utility of the treatment with ARNi has been assessed in preclinical models of various other CVDs. In some of the reviewed studies, ARNi administration was compared with ARB or ACEi monotherapy, allowing for several conclusions on the additive beneficial effect of concomitant administration of a NEPi. The study designs and their most important findings are summarized in Table 2 [102,103,104,105,106,107,108,109,110,111,112,113,114,115,116,117]. Additionally, the details of the experimental animals used, gavage protocols, and drug dosages are presented in Appendix A.

The majority of the available studies focused on investigating the effects of ARNi in different models of heart failure with preserved ejection fraction. Even though, as previously described, clinical trials have failed to demonstrate a statistically significant survival benefit of ARNi’s administration over valsartan monotherapy in patients with HFpEF, those preclinical studies described a vast range of mechanisms involved in ARNi’s cardioprotective effects and thus were included in this review. In those studies, HFpEF models were achieved either by the induction of pressure overload by aortic banding or transverse aortic constriction (TAC) [103,108,111,115,117]; by inducing hypertension (HT) with a high-salt diet [104] or Ang II infusion [116]; or by utilizing spontaneously hypertensive rats (SHRs) [105,112,114].

In a rat model of cardiac pressure overload, Lu et al. (2018) demonstrated that the administration of ARNi was superior to enalapril in preventing the occurrence of fibrotic myocardial depositions, cardiac hypertrophy, systolic myocardial dysfunction, and subsequent lung injury in the experimental animals [108]. Accordingly, the protein expressions of profibrotic (TGFβ; SMAD family member 3, Smad3), proapoptotic (mitochondrial Bax; cleaved caspase 3; poly(ADP-ribose)polymerase, PARP), and DNA-damage (phosphorylated H2A histone family member X, γ-H2AX) markers in the lung and LV myocardium, as well as the markers of pressure/volume overload (BNP; myosin heavy chain β, MHCβ), oxidative stress (NADPH oxidase 1, NADPH oxidase 2, oxidized protein), and mitochondrial damage (cytosolic cytochrome c) in LV myocardium were significantly reduced in animals receiving treatment with enalapril and further reduced after the treatment with ARNi. In a different study, by Burke et al. (2019), on HFpEF induced by pressure overload, the additive benefit of NEPi in the prevention of cardiac myofibroblast hyperplasia was demonstrated [103]. It was shown that treatment with ARNi altered the pathological state of cardiac fibroblasts by influencing their gene expression, leading them to quiescence rather than activation, which was not achieved by equimolar valsartan monotherapy. In their in vitro experiment on patient-derived cardiac fibroblasts, this effect was mediated by NPRA/PKG signaling, which led to the suppression of Ras homolog family member A (RhoA) activity by PKG-dependent phosphorylation. Antifibrotic effects of ARNi were also reported, by Suo et al. (2019), to be superior to ARB monotherapy in preventing left atrial fibrosis in the pressure overload mice model [115]. Interestingly, one study, by Norden et al. (2020), presented consistent results when considering the protective effects of ARNi treatment on cardiac hypertrophy and diastolic dysfunction in a pressure overload rat model, but without any significant effect on the extent of myocardial fibrosis in animals receiving either ARNi or valsartan monotherapy [111].

In a mice model of HFpEF, Ge et al. (2020) showed that ARNi administration significantly reduced LV dilation, improved LV systolic dysfunction, and alleviated cardiac hypertrophy and fibrosis [118]. Furthermore, this study included a robust assessment of lymphatic vasculature and inflammatory response in the myocardium after TAC, indicating an anti-inflammatory activity of ARNi. As in the aforementioned studies on MI models, it was shown that the administration of ARNi reduced the mRNA expression of several proinflammatory cytokines, namely IL6, IL1β, and TNFα, in the circulating blood. Moreover, ARNi decreased the protein expression of several factors associated with lymphangiogenesis: vascular endothelial growth factor C (VEGF-C), vascular endothelial growth factor receptor 3 (VEGFR3), and lymphatic vessel endothelial hyaluronan receptor 1 (LYVE-1) in cardiac tissue. As a result, the density of lymphatic vessels in the myocardium of ARNi-treated mice was reduced compared with the corresponding placebo groups, and a reduced accumulation of macrophages in the myocardium was observed. Similar protection against myocardial inflammatory activity by treatment with ARNi in the setting of pressure overload was reported by Li et al. (2020) [119]. It was shown that those beneficial anti-inflammatory effects were associated with the inhibition of the NF-κB-mediated NLRP3 inflammasome activation, leading to a decreased expression of profibrotic and proinflammatory cytokines. These findings on the inhibition of NLRP3 inflammasome by ARNi are in line with the previously described study in an MI model by Shen et al. (2021) [96].

Peng et al. (2020) demonstrated that ARNi ameliorated oxidative stress and exerted an antihypertrophic effect through the activation of sirtuin 3 in an AMP-activated protein kinase (AMPK)–dependent manner, leading to increased activity of a mitochondrial antioxidant enzyme: manganese superoxide dismutase (MnSOD) [120]. Sirtuin 3, a mitochondrial class III histone deacetylase, was previously shown to protect against cardiac hypertrophy and heart failure by preventing damage to cardiomyocyte mitochondria through several different pathways, including MnSOD activation [121,122].

Results obtained in the models of HFpEF related to longstanding hypertension are more varied and often conflicting. Kusaka et al. (2015) reported that treatment with ARNi decreased blood pressure, increased natriuresis, and decreased activity of the sympathetic nervous system in rats regardless of high-salt or low-salt conditions, and it did so more effectively than ARB alone [105]. Additionally, it was shown that in high-salt-loaded SHRs, treatment with ARNi ameliorated cardiac hypertrophy and inflammation, decreased the pathologic remodeling of the coronary arteries, and alleviated vascular endothelial dysfunction, in addition to what was achieved with valsartan monotherapy. On the contrary, Seki et al. (2017) showed that the addition of NEPi did not exert any additional benefit over valsartan monotherapy in ameliorating HT-related endothelial dysfunction [112].

Zhao et al. (2019) investigated in depth the effects of ARNi treatment on the RAAS system [117]. It was shown that the treatment with ARNi enhanced the activity of the protective arm of the RAAS, increasing the expression of ATR2 and MasR, which improved the imbalance of RAAS system in SHRs—which was not achieved in animals receiving valsartan only. This study showed that the cardioprotective effects of ARNi may be related to enhanced ATR2 expression and ACE2/Ang 1-7/MasR axis stimulation rather than to ATR1 inhibition. Similarly, in a mouse model of Ang II-induced cardiac hypertrophy, Tashiro et al. (2020) demonstrated that ARNi better prevented ventricular wall thickening and cardiomyocyte hypertrophy when compared with enalapril or valsartan [116]. Interestingly, this effect was uncoupled from cardiac fibrosis, as there were no differences in the extent of fibrosis and TGFβ mRNA expression among the control, valsartan, enalapril, and ARNi groups.

In a study by Sung et al. (2020), it was shown that high-dose ARNi treatment (dosage of 300 mg/kg/day) was superior to valsartan in reducing cardiac hypertrophy and lowering the susceptibility to ventricular arrhythmias [114]. ARNi had beneficial effects on cardiac electromechanical properties—it modulated repolarization dispersion and alleviated conduction heterogeneity. This finding was associated with the cardiac downregulation of the small-conductance calcium-activated potassium type 2 channel (SK channel) gene. Unexpected results were acquired by Hamano et al. (2019), who found that valsartan monotherapy, but not ARNi, initiated after 6 months of a high-salt diet in SHRs, improved cardiac hypertrophy and pulmonary edema, even though both drugs reduced systemic arterial blood pressure in a similar manner [104]. Even more surprisingly, ARNI increased the cardiac expression of the genes associated with hypertrophy and pulmonary edema (ANF; BNP; MHCβ; sarcoplasmic/endoplasmic reticulum Ca^2+^ ATPase 2a, SERCA2a) as compared with the placebo group. In the same study, results were antithetical when the treatment with ARNi or valsartan was initiated early in HT induction, where HT-related cardiac hypertrophy was not yet fully developed, underlining the crucial influence of the timing of the ARNi treatment initiation on its effectiveness.

Two animal studies implemented a preclinical model of cardiometabolic syndrome, investigating the effects of ARNi treatment on obesity-related cardiac diastolic dysfunction. Schauer et al. (2021) showed that obese rats treated with ARNi had improved diastolic dysfunction, which became significant after 4 weeks of treatment and continued to improve later on [123]. Additionally, ARNi significantly decreased LV collagen expression levels and ameliorated perivascular fibrosis, which was associated with the normalization of the phosphorylation levels of titin. Finally, treatment with ARNi slightly improved endothelial-dependent vasodilation in carotid arteries compared with the obese control group. Similarly, Aroor et al. (2021) showed that Zucker obese rats treated with ARNi were protected against diastolic and systolic myocardial dysfunction and from large artery stiffness, compared with valsartan or hydralazine monotherapy [102].

In a model of diabetic cardiomyopathy, Ge et al. (2019) showed that the administration of ARNi significantly attenuated cardiac remodeling and cardiac dysfunction related to diabetes mellitus and decreased oxidative stress levels and the expression of proinflammatory cytokines and proapoptotic factors [124]. Mechanistically, they showed that the observed anti-inflammatory, antioxygenation, antiapoptotic, and cardioprotective effects of ARNi were associated with the inhibition of the persistent phosphorylation of c-Jun N-terminal kinase (JNK) and p38 mitogen-activated protein kinase (p38MAPK), as well as the nuclear translocation of NF-κB factor. In a preclinical model of atrial fibrillation (AF), Li et al. (2021) found that modulating those signaling pathways may be also involved in the cardioprotective, antifibrotic, and antiarrhythmic effects of ARNi treatment [106]. In their study, ARNi was superior to valsartan monotherapy in attenuating atrial fibrosis and reducing AF inducibility, which was associated with the inhibition of the phosphorylation of Smad2/3, p38 MAPK, and JNK pathways, which was not observed in animals receiving valsartan alone.

Suematsu et al. (2018) investigated the effects of treatment with ARNi in a preclinical model of chronic kidney disease (CKD) [113]. Treatment with ARNi resulted in a substantial improvement in cardiac and aortic fibrosis, decreased activation of inflammatory and oxidative stress pathways in the myocardial tissue, and ameliorated LV mitochondrial mass loss, superior to valsartan monotherapy. It was shown that ARNi treatment suppressed the activation of NF-κB factor, inhibiting the activation of its downstream molecules, such as monocyte chemoattractant protein 1 (MCP-1) and NADP oxidase-4, preventing CKD-related cardiovascular dysfunction. In a preclinical model of autoimmune myocarditis, Liang et al. (2021) demonstrated that treatment with ARNi halted the damaging inflammatory response by inhibiting the differentiation of Th17 cells, probably via the sGC/NF-κB p65 signaling pathway [107].

Finally, there are two studies available that incorporated DOX administration to induce dilated cardiomyopathy as an element of cardiorenal syndrome in rats [125,126]. In both studies, in addition to a 5/6 nephrectomy, animals received an accumulated dosage of 7.5 mg/kg of DOX by four intraperitoneal injections at 5-day intervals within 20 days. First, Yang et al. (2019) described how a high-dose ARNi treatment (dosage of 100 mg/kg/day) significantly preserved residual renal and cardiac function and increased LVEF [125]. Additionally, ARNi attenuated renal and LV myocardial cellular injury reduced the extent of fibrosis and decreased the severity of oxidative stress levels in these two tissues. Moreover, ARNi administration was shown to significantly alleviate the increase in the renal expression of biomarkers for apoptosis (Bax, cleaved caspase 3, caspase 2, and caspase 9), fibrosis (Smad3 and TGFβ), mitochondrial damage (cytosolic cytochrome c), and autophagy. Accordingly, ARNi treatment prevented the decrease in the renal expression of antiapoptotic (Bcl-2, Bcl-XL) and antioxidant factors. In a follow-up study, Yeh et al. (2021) investigated the therapeutic impact of ARNi’s administration on the cardiomyocytes and cardiac function against oxidative stress and damage related to cardiorenal syndrome [126]. Once again, it was shown that therapy with ARNi significantly preserved LVEF and alleviated the fibrosis in the LV myocardium, along with reducing the hypertrophy of cardiomyocytes and decreasing the expression of BNP and the proinflammatory cytokine interleukin 33. Additionally, in animals receiving treatment with ARNi, the protein expressions of markers of oxidative stress (NADPH oxidase 1; NADPH oxidase 2; p22^phox^; oxidized protein), fibrosis (Smad-3; TGFβ), apoptosis (mitochondrial Bax; caspase-3; PARP), mitochondrial damage (dynamin-related protein 1, Drp1; cyclophilin-D, cytosolic cytochrome c), and autophagy (beclin 1; autophagy related 5, Atg5) were all decreased. What is more, the described study included an in vitro investigation of the effects of ARNi administration on functional and morphological mitochondrial disturbances in cardiorenal syndrome. It was demonstrated that ARNi treatment resulted in a significant decrease in H_2_O_2_-induced apoptosis, the severity of oxidative stress, autophagy, and mitochondrial damage, through the inhibition of Mitofusin-2 (Mfn2) activity, therefore preventing the fusion of damaged mitochondria with the healthy mitochondrial population [126].

Studies of ARNi administration in animal models of different CVD have shown that it provides functional improvement and molecular cardioprotection by decreasing oxidative stress levels and endothelial dysfunction; increasing antioxidant activity; inhibiting proinflammatory pathways, especially related to NF-κB factor; providing protection against mitochondrial damage; and modulating the imbalance of RAAS—increasing the activity of cardioprotective pathways related to ATR2 and MasR. Consistent with studies in MI and HFrEF models, ARNi administration was described as directly targeting different processes parallel to those involved in AIC pathophysiology. Thus, there is convincing evidence available that ARNi administration should be robustly studied as a potential preventive measure in the setting of DOX administration.

## 8. ARNi in Preclinical Models of AIC

Although, as described above, ARNi administration has been extensively studied in various models of CVDs, its effectiveness in animal models of AIC remains fairly understudied. At the time of writing, there have been seven preclinical studies and a conference report on this topic published. The animal models used, experimental groups, and the most important findings of those studies are summarized in Table 3 [48,127,128,129,130,131,132,133]. Additionally, details on experimental animals, the protocols of DOX administration, and oral drug gavage are provided in Appendix A.

The first report on the effects of ARNi administration on AIC in an animal model was published by Xia et al. (2017) [131]. In that study, performed on a mice model receiving high doses of intraperitoneal DOX for 2 weeks, ARNi administration was shown to significantly attenuate LVEF decrease and alleviate cardiac hypertrophy and fibrosis. Additionally, ARNi decreased cardiomyocyte apoptosis and improved single cardiomyocyte contractile function. A detailed analysis of mitochondrial morphology by using electron microscopy showed that DOX-related pathologic changes to the mitochondrial shape, associated with decreased activity of respiratory enzymes, were alleviated by concomitant ARNi administration. In a subsequent in vitro experiment, this effect of ARNi on mitochondrial morphology was linked to decreased phosphorylation of Drp1, describing a novel mechanism of ARNi cardioprotective effects. This was further studied by Boutagy et al. (2020) on a rat model of chronic AIC, with animals receiving low doses of intraperitoneal DOX for 3 weeks and concomitant oral placebo, valsartan or ARNi for 6 weeks [127]. The administration of ARNi or valsartan showed similar efficacy in preventing histological evidence of cellular damage (changes in myofiber variation or vacuolation). However, only ARNi administration prevented the decline in LVEF, as opposed to rats receiving valsartan monotherapy, in which LVEF did not differ from the placebo-receiving group. Additionally, by measuring the uptake level of a radiolabeled tracer that binds to the active catalytic site of several MMPs, namely MMP-2, MMP-3, MMP-7, MMP-9, MMP-12, and MMP-13, it was shown that in ARNi-receiving rats, there was lower myocardial MMP activity.

Contrary to the described studies, Miyoshi et al. (2022) found no differences in the echocardiographic assessment of LV systolic dysfunction between rats receiving DOX with concomitant oral saline, valsartan, and ARNI [130]. However, they observed the significant attenuation of cardiac troponin C and N-terminal prohormone of brain natriuretic peptide (NT-proBNP) levels, cardiomyocyte hypertrophy, and cardiac fibrosis in the ARNi-treated animals. This was associated with decreased levels of ROS generation and decreased oxidative stress levels, which were further confirmed in an in vitro experiment. Additionally, ARNi treatment ameliorated the decrease of antiapoptotic protein Bcl-2 levels in the myocardium and prevented a DOX-induced decrease in the phosphorylation of AMPK. Two animal studies assessed the effects of ARNI administration on DOX-related ECG changes [128,132]. Proarrhythmogenic changes in ECG associated with DOX administration, such as prolongation of QRS duration, ST interval, and an increase in QT/PQ index, were alleviated by ARNi administration in both articles, showing the potential of this drug combination in preventing ventricular arrhythmias. Additionally, in both papers, the influence of ARNi on oxidative stress levels was assessed. Yu et al. (2021) showed that ARNi administration was associated with increased activity in antioxidant enzymes, such as superoxide dismutase and catalase, and decreased lipid peroxidation in both ARNi groups compared with the animals receiving placebo [132]. Similarly, Dindas et al. (2021) observed a decrease in total oxidant status and an increase in total antioxidant status in DOX-treated mice receiving ARNi compared with those receiving placebo [128]. Additionally, they showed a decrease in the serum levels of inflammatory cytokines, namely TNFα, IL1β, and IL6, in the ARNi-treated group.

Ye et al. (2021) have shown that the beneficial effects of ARNi administration in the mice model of chronic AIC were parallel to those observed in mice with knockout of the TLR-2 gene [48]. A subsequent in vitro experiment showed that ARNi disrupted the formation of a TLR2- MyD88 complex induced by DOX, inhibiting the TLR-2/MyD88/NF-kB downstream pathway and preventing an inflammatory response [48]. In a conference report by Maurea et al. (2022), it was shown that treatment with ARNi alleviated the DOX-induced increase in the cardiac activity of this pathway, with a decreased cardiac expression of NLRP3 inflammasome, MyD88, damage-associated molecular patterns (DAMPs) and NF-kB in ARNi-treated mice [133]. Those findings were further supported by a recent conference report in which in an in vitro experiment on human cardiomyocytes treated with DOX or trastuzumab, ARNi was shown to decrease the activity of the MyD88/NF-kB/NLRP3 pathway, with a synergistic effect of cotreatment with an SGLT-2 inhibitor dapagliflozin [134].

Finally, Kim et al. (2022) associated the cardioprotective effects of ARNi in the AIC model with decreased levels of endoplasmatic reticulum stress, as it was shown to inhibit the expression of endoplasmatic reticulum stress markers (glucose-regulated protein 78, GRP78; protein kinase RNA-like endoplasmic reticulum kinase, PERK; inositol-requiring transmembrane kinase endoribonuclease-1α, IRE-1α; activating transcription factor 4, ATF-4; activating transcription factor 6, ATF-6; activated eukaryotic initiation factor 2 alpha, eIF-2α; and CCAAT-enhancer-binding protein homologous protein, CHOP) in the myocardium of a rat model [129].

## 9. ARNi in Human Studies on AIC

In a recent prospective study, ARNi was shown to be safe and effective in patients with CTRCD thanks to previous chemotherapy for breast cancer (mainly anthracyclines and anti–human epidermal growth factor receptor 2 (HER2) treatment) [9]. Additionally, one case report of ARNI administration as a first-line treatment for CTRCD is available, in which the importance of the proper management of ARNI’s hypotensive effect is underlined [135]. Additionally, in a form of a short letter to the editor, Martin-Garcia et al. (2019) presented how ARNi administration in 10 patients with CTRCD experienced significant improvement in LVEF visualized by CMR [136]. At the time of writing, no human-derived data are available on ARNi in AIC primary prevention. However, there is currently an ongoing multicenter, randomized, placebo-controlled, double-blind phase-2 clinical trial (Prevention of Cardiac Dysfunction During Breast Cancer Therapy; PRADAII) aimed at assessing the efficacy of ANRi during anthracycline-containing chemotherapy for breast cancer in the primary prevention of CTRCD. At the time of writing, the PRADAII trial is still recruiting to reach the estimated enrollment of 214 patients. The trial is expected to be completed by late 2025.

## 10. Summary and Future Perspectives

In recent years there has been growing, substantial interest in the field of cardio-oncology. It is widely recognized that its development is crucial for ensuring the well-informed, evidence-based management of potential cardiovascular risks related to oncologic therapy. Accordingly, the development of new and improved methods for CTRCD primary prevention is required, as HF related to chemotherapy, especially anthracycline-based, still poses a significant clinical challenge.

As described, robust data are available on the cardioprotective effects of ARNi administration in various preclinical models of CVDs. In animal models of MI, ARNi attenuates the immediate decrease in LVEF and prevents the development of post-MI HFrEF by alleviating myocardial fibrosis and hypertrophy, decreasing oxidative stress levels and an excessive inflammatory response. Studies on models of other CVDs, especially HFpEF, have also demonstrated ARNi’s beneficial effects on endothelial dysfunction, mitochondrial damage, and the imbalance of RAAS. As there are many parallels in the pathophysiology of the modeled diseases and the mechanisms of the development of AIC, this makes ARNi a relevant candidate for detailed research assessing its potential effectiveness in the primary prevention of AIC. Additionally, most of the reviewed studies reported that this effect was more pronounced in ARNi-receiving animals than in those receiving monotherapy with ARBs or ACEi.

Currently, data on ARNi administration in AIC prevention remain limited to several preclinical studies, which have shown its potential to ameliorate systolic dysfunction, decrease cellular damage and myocardial fibrosis, and prevent proarrhythmic ECG changes associated with DOX exposure. The beneficial cardioprotective effects of ARNi administration in the presented studies of AIC prevention included a decrease in the levels of oxidative stress and proinflammatory cytokines, a decrease in the myocardial inflammatory response, the prevention of morphologic and functional changes to cardiomyocyte’s mitochondria, the modulation of matrix metalloproteinases activity, and the amelioration of endoplasmic reticulum stress. At the time of writing, only two studies of ARNi administration in AIC preclinical models have demonstrated the added benefit of ARNi administration over ARB monotherapy, as other studies lacked the comparator group receiving valsartan in monotherapy. However, the beneficial effect of adding NEPi to the regimen has already been extensively described in animal models of other CVDs and demonstrated in the PARADIGM-HF trial in HFrEF patients. Additionally, the described preclinical studies of AIC varied in terms of DOX administration protocols, as well as time of onset and length of oral drugs’ gavage. As all of the studies demonstrated some benefits of ARNi coadministration during DOX treatment, the question of when the oral treatment should be initiated to provide optimal results remains unanswered. Even though primary data are available to support the beneficial effects of NEP inhibition in AIC, changes to NEP expression and activity during anthracycline treatment, as well as the effects of NEPi monotherapy in this setting, have not been studied. Clinical data on ARNi in patients with HF from AIC remain scarce, and there are no human-derived data on their effectiveness in AIC primary prevention. At the time of writing, the PRADAII trial assessing the effectiveness of ARNi coadministration during chemotherapy in breast cancer patients is still ongoing, with the results to be published in several years. It is important to underline here that one of the potential limitations of ARNi administration in primary the prevention of AIC is resultant hypotension. A significant portion of patients undergoing anticancer treatment have low or low-normal blood pressure, which can be associated with their nutritional status, decreased thirst, and poor general health status. As there is considerable risk that those patients might not tolerate ARNi treatment, clinical trials addressing this question are necessary. Thus, the exact role of NEP and its inhibition in the development and prevention of AIC is yet to be understood.

## 11. Conclusions

As demonstrated in preclinical studies on a wide variety of cardiovascular diseases, the cardioprotective effects of ARNi administration are associated with decreased oxidative stress levels, the inhibition of myocardial inflammatory response, protection against mitochondrial damage and endothelial dysfunction, and improvement in the RAAS imbalance. However, data on ARNi’s effectiveness in the prevention of AIC remain limited. Studies of ARNi administration in animal models of AIC show promising results, as ARNi treatment prevented DOX-induced ventricular systolic dysfunction and electrocardiographic changes and ameliorated oxidative stress, mitochondrial dysfunction, endoplasmic reticulum stress, and the inflammatory response. In a few of those studies, ARNi proved more effective than ARB monotherapy in AIC prevention. Human data remain limited to a few reports on ARNi administration in patients in whom AIC had already developed. At the time of writing, there is currently an ongoing PRADAII trial aiming to assess the efficacy of ARNi in patients receiving anthracycline-based chemotherapy for breast cancer, expected to be completed by late 2025.

## Figures and Tables

**Figure 1 cancers-15-00312-f001:**
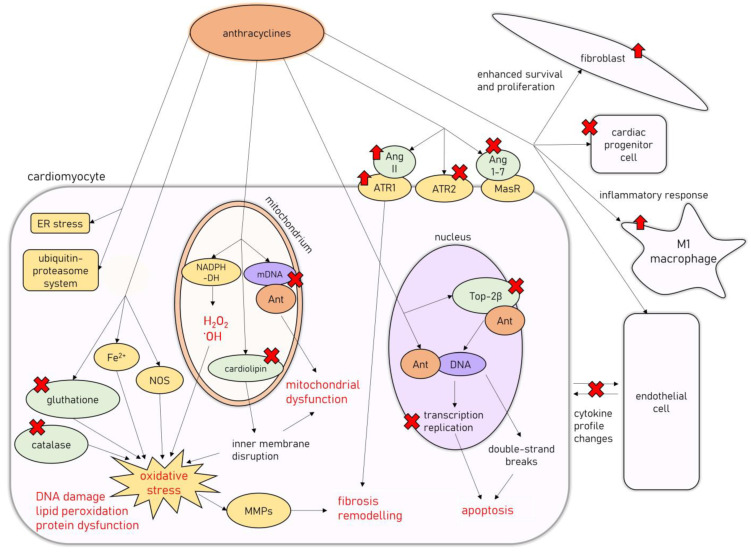
Molecular mechanisms of AIC. Abbreviations: Ant—anthracycline; Ang 1-7, angiotensin-1-7; Ang II—angiotensin II; ATR1—angiotensin II receptor type 1; ATR2—angiotensin II receptor type 2; DNA—deoxyribonucleic acid; mDNA—mitochondrial DNA; ER—endoplasmic reticulum; MasR—Mas receptor; MMPs—metalloproteinases; NADPH-DH—nicotinamide adenine dinucleotide phosphate dehydrogenase; NOS—nitric oxide synthase; Top2β—topoisomerase 2β.

**Figure 2 cancers-15-00312-f002:**
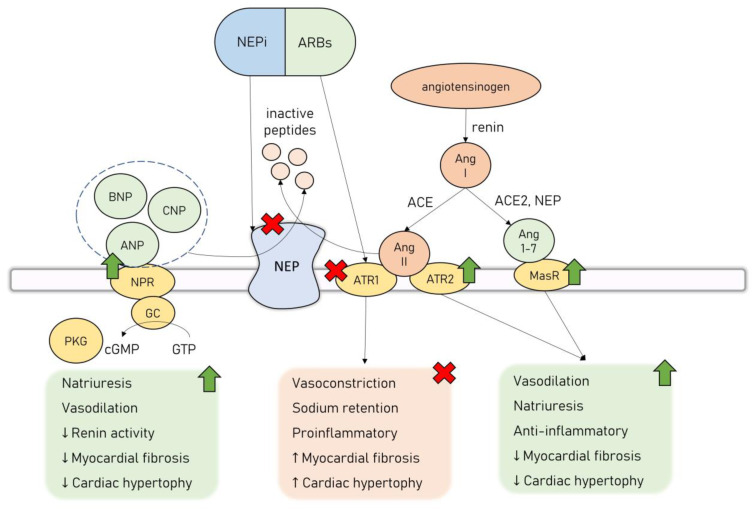
Dual neprilysin and angiotensin II receptor type 2 blockage by ARNi. Abbreviations: ACE—angiotensin-converting enzyme; Ang 1-7—angiotensin 1-7; Ang II—angiotensin II; ANP—atrial natriuretic peptide; ARB—angiotensin II receptor type 1 blocker; ATR1—angiotensin II receptor type 1; ATR2—angiotensin II receptor type 2; BNP—brain natriuretic peptide; cGMP—cyclic guanosine monophosphate; CNP—C-type natriuretic peptide; GC—guanyl cyclase; GTP—guanosine-5′-triphosphate; MasR—Mas receptor; NEP—neprilysin; NEPi—neprilysin inhibitor; NPR—natriuretic peptide receptor; PKG—protein kinase G.

**Table 1 cancers-15-00312-t001:** ARNi administration in preclinical models of MI.

Paper by	Animal Model	Gavage Initiation	Groups	LVEF (%)		Other Findings
Chang [89]	SPRD rats	1 wk after MI	Vehicle Enalapril ARNi	38.5 ± 2.0 46.7 ± 9.1 ↑ 57.6 ± 5.5 ↑↑		↓↓ HW/BW ratio in ARNi ↓↓ Ventricular arrhythmias inducibility in ARNi ↑↑ Expression of K^+^ channel proteins in ARNi
Chang [90]	New Zealand White rabbits	1 wk after MI	Vehicle ARB ARNi	37.1 ± 6.3 44.3 ± 6.3 ↑ 53.8 ± 10.0 ↑↑		↓↓ Ventricular arrhythmia inducibility in ARNi
Ishii [91]	Mice	1 day after MI	Vehicle Enalapril ARNi	NA		↓↓ Post-MI mortality rate due to LV rupture in ARNi ↑↑ %FS 14 and 28 days post-MI in ARNi ↓↓ Myocardial expression of IL1β, IL6, and MMP-9 mRNA in ARNi No differences in myocardial fibrosis and inflammatory infiltration.
Kompa [92]	SPRD rats	1 wk after MI	Vehicle Perindopril ARNi	40.46 ± 1.27 42.22 ± 1.16 46.65 ± 0.83 ↑↑		ARNi improved end-systolic pressure-volume relationship compared with perindopril ↓ LV mass, cardiomyocyte CSA, and cardiac fibrosis in perindopril and ARNi ↓↓ ANP, MHC β, and TIMP2 expression in ARNi
Liu [93]	C57BL/6J mice	Directly after MI	Vehicle Benazepril ARNi ARNi + Benazepril	= 58.7 ± 0.42 ↑ 62.35 ± 0.25 ↑↑		HW/BW ratio ↓ in ARNi and benazepril and ↓↓ in ARNi + benazepril Myocardial fibrosis ↓ in ARNi and ↓↓ in ARNi + benazepril ↓↓ TGFβ1 expression in ARNi and ARNi + benazepril No differences in IL6 and TNFɑ expression.
Pfau [94]	Lewis rats	1 wk after MI	Vehicle ARB ARNi	1 wk: 34 ± 2 38 ± 2 39 ± 2 ↑	5 wks: 35 ± 2 39 ± 2 42 ± 2	↓ HW/TL ratio and fibrosis in ARNi ↓ Myocyte CSA in ARNi and ARB ↓ Expression of CTGF, MHCβ, MHCβ/α, and ANP in ARNi and ARB
Raj [95]	SPRD rats	Directly after MI	Vehicle ARB ARNi	56.60 ± 1.70 65.45 ± 2.70 ↑ 66.82 ± 1.43 ↑		↓ Oxidative stress in ARNi and ARB ↓ TNFα, collagen, and BNP in ARNi and ARB
Shen [96]	SPRD rats	1 wk after MI	Vehicle ARNi	3 days: ↑	7 days: ↑	↓ Interstitial fibrosis in ARNi ↓ Serum IL1βa and IL18 levels in ARNi ↓ ROS and NLRP3 inflammasome activation in ARNi
Suematsu [97]	C57BL/6J diabetic mice	Day after MI	Vehicle ARB ARNi	29 ± 3.2 = 43 ± 3.4 ↑		↓↓ LV fibrosis and expression of TGFβ mRNA in ARNi ↓ HW/BW ratio in ARB and ARNi ↓ ANP mRNA in ARNi
Torrado [98]	New Zealand White rabbits	At reperfusion only	Vehicle ARB ARNi	ARB = ARNi ↑		↓ Infarct size in ARB and ARNi and ↓ cardiac troponin I serum concentration in ARNi
		At LVEF ≤ 40%		ARB = ARNi ↑↑		
		At reperfusion		4 wks: ARB = ARNi ↑↑	10 wks: = ↑	↓ Infarct size in ARNi
Trivedi [99]	SHRs	4 wks after reperfusion	Vehicle ARB ARNi	= = ↑↑		↓ Infarct border zone expansion in ARB and ARNi Aortic vasorelaxation responses to Ach and SNP ↑ in ARB and ↑↑ in ARNi ↑↑ Myocardial NO bioavailability in ARNi No differences in fibrosis between groups
Vaskova [100]	SPRD rats	1 wk after MI	Vehicle ARB ARNi	36.79 ± 2.1 40.68 ± 4.8 ↑ 41.42 ± 3.4 ↑		↑ Production of plasma exosomes in ARB and ARNi ↓↓ Expression of rno-miR-181a in ARNi ↓ Fibrosis in ARB and ARNi
Von Lueder [101]	Lewis rats	1 wk after MI	Vehicle ARNi	47 ± 5 60 ± 2 ↑		Improved LV function in pressure-volume loops in ARNi ↓ LV mass and fibrosis in peri-infarct and remote myocardium in ARNi No differences in infarct size and perivascular fibrosis.

Abbreviations: Ach—acetylcholine; ANP—atrial natriuretic peptide; ARNi—sacubitril/valsartan, ARB—valsartan; BW—body weight; cont.—continued; CSA—cross-sectional area; FS—fractional shortening; HW—heart weight; IL—interleukin; LV—left ventricle; LVEF—left ventricle ejection fraction; MHCβ—myosin heavy chain β; MI—myocardial infarction; MMP—matrix metalloproteinase; mo—month; NA—not assessed; NLRP3—NLR family pyrin domain containing 3; NO—nitric oxide; ROS—reactive oxygen species; SNP—sodium nitroprusside; SPRD—Sprague Dawley rats; TGFβ—transforming growth factor β; TL—tibial length; TNFα—tumor necrosis factor α; wk—week. Symbols: ↑/↓—significantly increased/decreased compared with placebo; ↑↑/↓↓—significantly increased/decreased compared with placebo and other groups; =—no significant change compared with placebo. Groups and results presented in Table 1 were chosen because of their importance for the review, but they do not exhaust all of the results presented in the selected papers.

**Table 2 cancers-15-00312-t002:** ARNi administration compared with ARB or ACEi monotherapy in preclinical models of other cardiovascular diseases.

Model of	Paper by	Species	Gavage Initiation	Groups	Findings *
HFpEF due to pressure overload	Burke [103]	C57Bl6/J mice	A day before TAC, cont. for 4 wks	Vehicle ARB ARNi	↑↑ LVEF in ARNi ↓↓ Interstitial fibrosis and fibroblast population in ARNi ↓↓ Cardiomyocyte CSA and HW/TL in ARNi
	Lu [108]	SPRD rats	4 wks after TAC, cont. for 32 days	Vehicle Enalapril ARNi	↓↓ Sarcomere-length, left ventricle fibrotic area, cardiomyocyte size and lung injury in ARNi ↓↓ Expressions of fibrotic, oxidative, apoptotic, DNA damage, mitochondrial damage, volume overload markers in LV in ARNi
	Norden [111]	SPRD rats	Cont. for 8 wks	Vehicle ARB ARNi	↓↓ LV weight in ARNi ↓↓ Diastolic dysfunction in ARNi No differences in LVEF and myocardial fibrosis
	Suo [115]	C57BL/6J mice	8 wks after TAC, cont. for 4 wks	Vehicle ARB ARNi	↑↑ LVEF in ARNi ↓↓ Fibrosis in ARNi
HT	Hamano [104]	SHRcp fed high-salt diet	I: 6 mos, with high-salt diet	Vehicle ARB ARNi	↓ LV/BW ratio in ARNi in Plan I ↓↓ LV/BW and pulmonary edema in ARB in Plan II No differences in cardiomyocyte CSA and fibrosis
			II: After 6 mos of high-salt diet, cont. for 6 mos		
	Kusaka [105]	SHRcp fed high-salt diet	Cont. for 4 wks	Vehicle ARB ARNi	↓ LV in ARNi ↓↓ Myocardial fibrosis in ARNi ↓↓ Impairment of acetylcholine-induced vascular relaxation in ARNi
	Seki [112]	SHRs	Cont. for 12 wks	Vehicle ARB ARNi	Endothelium-dependent hyperpolarization-mediated responses improved similarly in ARB and ARNi ↓ LV in ARNi
	Sung [114]	SHRs	Cont. for 2 wks	Vehicle ARB ARNi	↓↓ Diastolic dysfunction and ↓↓ ventricular hypertrophy in ARNi ↓↓ Incidence of ventricular arrhythmias in ARNi No differences in LVEF
	Tashiro [116]	C57BL/6J mice	Started on the 7th day of Ang II infusion, cont. for 2 wks	Vehicle Enalapril ARB ARNi	↓↓ LV concentric hypertrophy in ARNi Myocyte CSA ↓ in ARB, ↓ in enalapril, and ↓↓ in ARNi No differences in fibrosis and TGFβ expression
	Zhao [117]	SHRs	Cont. for 12 wks	Vehicle ARB ARNi	LVEF ↑↑ in ARNi and ↑ in ARB ↓↓ LV mass in ARNi ↓ Fibrosis, TGFβ expression and nNOS, eNOS protein expression in ARB and ARNi ↓↓ ACE, ATR1, and ↑↑ ACE2, MasR, ATR2 cardiac protein expression in ARNi
HF due to volume overload (by AVI)	Maslow [109]	SPRD rats	4 wks after AVI, cont. for 4 wks	Vehicle ARB NEPi ARNi	Improved load-dependent indexes of left ventricle contractility and relaxation only in ARNi Improved load-independent index of contractility in ARB and ARNi ↑↑ Exercise tolerance in ARNi ↓↓ Myocardial fibrosis in ARNi
	Maslow [110]	SPRD rats	On the day of AVI, cont. for 8 wks	Vehicle ARB NEPi ARNi	↑ LVEF in ARNi ↓ Myocardial fibrosis in ARB, NEPi, and ARNi ↑ Exercise tolerance in ARB and ARNi
HFpEF due to obesity	Aroor [102]	Zucker Obese rats	At 16 wks of age, cont. for 10 wks	Vehicle ARB ARNi	↑ LVEF, ↓ fibrosis, and ↓ oxidative stress in ARB and ARNi ↑ Endothelial-dependent aortic relaxation in ARB and ARNi ↑↑ E’/a’ ratio in ARNi
AF	Li [106]	SPRD rats	After AF induction, cont. for 4 wks	Vehicle ARB ARNi	↑ LVEF in ARB and ARNi ↓↓ Atrial fibrosis and susceptibility to AF in ARNi
Myocarditis	Liang [107]	BALB/c mice	On the day of myocarditis, cont. for 3 wks	Vehicle ARB ARNi	↓↓ HW/BW ratio, ↓↓ myocardial histopathologic scores, and ↓↓ cTnT levels in ARNi ↓↓ Serum hsCRP, IL6, and serum/myocardial IL17 levels in ARNi ↓↓ Th17 cells and their transcription factors in myocardial tissue in ARNi
CKD	Suematsu [113]	SPRD rats	2 wks after nephrectomy, cont. for 8 wks	Vehicle ARB ARNi	HW/BW ratio, myocyte CSA, markers of oxidative stress, myocardial and aortic fibrosis ↓↓ in ARNi and ↓ in ARB ↓↓ expression of NF-κB, COX-2 in ARNi

Abbreviations: Ach—acetylcholine; AF—atrial fibrosis; ARNi—sacubitril/valsartan; ARB—valsartan; AVI—aortic valve insufficiency; BW—body weight; CSA—cross-sectional area; CKD—chronic kidney disease; cont.—continued; HFpEF—heart failure with preserved ejection fraction; HT—hypertension; HW—heart weight; LV—left ventricle; LVEF—left ventricle ejection fraction; mo—month; NEPi—sacubitril; TAC—transverse aortic constriction; TL—tibial length; SHRs—spontaneously hypertensive rats; SPRD—Sprague Dawley; wk—week. Symbols: ↑/↓—significantly increased/decreased compared with placebo; ↑↑/↓↓—significantly increased/decreased compared with placebo and other treatments. * Groups and results presented in Table 2 were chosen because of their importance for the review, but they do not exhaust all of the results presented in selected papers.

**Table 3 cancers-15-00312-t003:** ARNi administration in animal models of anthracycline-induced cardiotoxicity.

Paper by	Animal Model	Groups + Dosage (mg/kg/d)	Other Findings (Presented in Comparison to DOX + Vehicle Groups)
Boutagy [127]	Wistar rats	DOX + Vehicle DOX + ARB 31 DOX + ARNi 68	↑ LVEF in ARNi ↓ Myocyte vacuolation in ARNi and ARB ↓ Myocardial fibrosis in ARNi and ARB at 4 wks (no longer seen at 6 wks) ↓ Capillary density in ARNi at 6 wks ↓ Matrix metalloproteinases activity in ARNi at 4 wks ↓ Myocyte CSA and heart weight in all DOX-receiving groups No differences in cellular apoptosis between groups
Dindas [128]	Balb-c mice	Vehicle ARNi 80 DOX + Vehicle DOX + ARNi 80	↓ Degenerative changes and streaking in cardiomyocytes in DOX + ARNi ↓ QRS duration, ST interval and QT/PQ index in DOX + ARNi ↓ NT-proBNP, TNFα, IL1β, IL6, and caspase 3 in DOX + ARNi ↓ Total oxidant status and ↑ total antioxidant status in DOX + ARNi
Kim [129]	Sprague Dawley rats	Vehicle DOX + Vehicle DOX + ARNi 60	↓ Cardiomyocyte apoptosis in ARNi ↓ Endoplasmic reticulum stress in ARNi ↓ Serum cardiac troponin I and NT-proBNP levels in ARNi group
Maurea [133]	C57Bl/6 mice	Sham Sac/Val 60 DOX DOX + Sac/Val 60	ARNi improved EF and prevented the reduction of radial and longitudinal strain ↓ Cardiac expression of NLRP3, MyD88, DAMPs, and NF-kB in ARNi ↑ Expression of phosphorylated AMPK in ARNi ↓ Levels of Calgranulin S100 and galectine-3 in ARNi
Miyoshi [130]	Sprague Dawley rats	DOX + Vehicle DOX + Val 31 DOX + ARNi 68	No differences in LVEF and FS between groups. ↑ Cardiomyocyte CSA and ↑ cardiac fibrosis in ARNi ↑ Cardiac TNFα and ANP mRNA expression in ARB and ARNi ↓ Myocardial collagen I mRNA expression in ARNi ↓ Cardiac troponin T and NT-proBNP levels in ARNi ↓ Cardiac reactive oxygen species levels in ARNi ↑ Phosphorylation of AMPK and ↑ Bax/Bcl-2 ratio in ARNi
Xia [131]	Balb-c mice	Vehicle DOX + Vehicle DOX + ARNi 80	↓ Cardiac hypertrophy, myocardial fibrosis and cellular apoptosis in ARNi ↓ Heart weight/body weight and ↑ heart weight/tibial length in ARNi ↑ Single cardiomyocyte contractile function in ARNi ↓ Pathologic changes to mitochondria and ↓ Drp1 expression in ARNi ↓ Cleaved caspase 3 in ARNi
Ye [48]	C57BL/6 mice	Vehicle DOX + Vehicle DOX + ARNi 60 TLR2 KO + Vehicle TLR2 KO + DOX	↑ LVEF in ARNi, TLR2 KO, and TLR2 KO + DOX ↓ Ventricular wall thinning and ↓ heart cavity enlargement in ARNi and TLR2 KO ↓ Myocardial fibrosis in ARNi and TLR2 KO ↓ Myocardial collagen I and TGFβ protein levels in ARNi and TLR2 KO ↓ Myocardial TNFα and NF-κB levels in ARNi and TLR2 KO
Yu [132]	New Zealand white rabbits	Vehicle DOX + Vehicle DOX + ARNi 5 DOX + ARNi 10	↓ PR segment, QRC segment prolongation, QT interval and QT/PQ index in both ARNi groups ↓ Serum BNP level in ARNi groups ↑ Activity of superoxide dismutase and catalase and ↓ lipid peroxidation in both ARNi groups

Abbreviations: AMPK—adenosine monophosphate-activated protein kinase; ANP—atrial natriuretic peptide; ARB—valsartan; ARNi—sacubitril/valsartan; BNP—brain natriuretic peptide; CSA—cross-sectional area; DOX—doxorubicin; Drp1—dynamin-related protein 1; KO—knockout; IL—interleukin; LVEF—left ventricle ejection fraction; NF-κB—nuclear-factor kappa-light-chain-enhancer of activated B cells; NLRP3—NLR family pyrin domain containing 3; NT-proBNP—N-terminal prohormone of brain natriuretic peptide; MyD88—myeloid differentiation primary response 88; TGFβ—transforming growth factor β; TNFα—tumor necrosis factor α; wks—weeks. Symbols: ↑/↓—significantly increased/decreased.

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
