# Peer review of "Neprilysin Inhibition in the Prevention of Anthracycline-Induced Cardiotoxicity"

_cancers, 2023, doi:10.3390/cancers15010312_

Round 1

Reviewer 1 Report

Sobiborowicz-Sadowska and colleagues present a comprehensive review covering anthracycline cardiotoxicity and neprilysin inhibition at the preclinical and clinical levels, providing a rationale for pursuing neprilysin inhibition for the prevention of anthracycline-induced cardiotoxicity. The authors are to be commended for covering these topics in exquisite detail in a manner that is easy for the reader to follow. The Tables are also very well constructed and easy to interpret.

Major comments:

1.       Page 6, line 227 – As part of the discussion on dexrazoxane, it is important to mention that one of two randomized breast cancer trials found reduced tumor response to anthracyclines when coadministered with dexrazoxane, without change in overall survival or progression free survival (PMID 9193323). This has led to decreased use of dexrazoxane in adult oncology practice, even though subsequent expert statements and meta-analysis (PMID 34396164, 14564513, 33766818) have emphasized no difference in anti-tumor efficacy with dexrazoxane.

2.       Page 6, line 241 – At the level of clinical studies, it is not clear that RAAS inhibition has a definite cardioprotective effect for prevention of anthracycline-induced cardiotoxicity. In the PRADA study (PMID 26903532), there was lower rate of LVEF decline on candesartan vs. placebo; however in the OVERCOME trial (PMID 23583763), there was no difference in rate of LVEF decline on enalapril+carvedilol vs placebo. This may be related to differences in baseline CV risk between study populations. In general, the cardio-oncology uses BB/RAAS inhibition for cardioprotection in high risk situations but acknowledges that the data is somewhat contradictory.

3.       Page 8, line 339 – this section on ARNi in preclinical models of MI can be shortened, since it is of lesser relevance to anthracycline induced cardiotoxicity. This section is actually very nicely summarized in Supplemental Table 1 – consider moving Supplemental Table 1 into the main manuscript, and shortening the text significantly.

4.       Page 11, line 482 – the clinical relevance of the preclinical studies on HFpEF are unclear, since the PARAGON-HF trial showed no clinical benefit to angiotensin-neprilysin inhibition in HFpEF. Consider shortening the text of this section and moving Supplemental Table 2 into the main text, as in comment #3 above

5.       Page 16, line 735 – in this section on ARNi in human studies on AIC, consider mentioning that hypotension may be a limiting factor in the use of ARNi in the cardio-oncology population. Many cardio-oncology patients have low-normal blood pressures either related to poor oral intake or general deconditioning, and may not tolerate the potent antihypertensive effects of ARNi.

Author Response

Dear Reviewer,

Ref.: cancers-2093839
Neprilysin inhibition in the prevention of anthracycline-induced cardiotoxicity.

We would like to thank you for the careful and thorough reading of this manuscript and for the thoughtful comments and constructive suggestions, which help to improve the quality of this manuscript. We hope that the changes we included will be acceptable and will make our manuscript suitable for publication in Cancers. All the authors have read and approved the final manuscript.

Below we provide our responses to each of the Reviewer’s comments:

1. Page 6, line 227 – As part of the discussion on dexrazoxane, it is important to mention that one of two randomized breast cancer trials found reduced tumor response to anthracyclines when coadministered with dexrazoxane, without change in overall survival or progression free survival (PMID 9193323). This has led to decreased use of dexrazoxane in adult oncology practice, even though subsequent expert statements and meta-analysis (PMID 34396164, 14564513, 33766818) have emphasized no difference in anti-tumor efficacy with dexrazoxane.

Thank you for your important suggestion, providing valuable insight into potential doubts associated with the use of dexrazoxane in clinical practice. Thus, we have written:

“However, one of the randomized clinical trials of dexrazoxane vs. placebo in breast cancer patients receiving anthracycline-based chemotherapy, reported decreased tumor objective response rates in the dexrazoxane-receiving group (46.8% for dexrazoxane and 60.5% for placebo, 95% confidence interval: -25% to -2%; p=0.019), without any changes to overall survival and progression-free survival. Though this finding was not supported by several meta-analyses and expert statements, underlining no change in the anti-tumor efficacy of anthracyclines when co-administered with dexrazoxane, it was associated with decreased enthusiasm for the use of dexrazoxane in adult cancer patients.”

2. Page 6, line 241 – At the level of clinical studies, it is not clear that RAAS inhibition has a definite cardioprotective effect for prevention of anthracycline-induced cardiotoxicity. In the PRADA study (PMID 26903532), there was lower rate of LVEF decline on candesartan vs. placebo; however in the OVERCOME trial (PMID 23583763), there was no difference in rate of LVEF decline on enalapril+carvedilol vs placebo. This may be related to differences in baseline CV risk between study populations. In general, the cardio-oncology uses BB/RAAS inhibition for cardioprotection in high risk situations but acknowledges that the data is somewhat contradictory.

Thank you for the comment. We have incorporated this, highlighting the uncertainty in the manuscript:

“The PRADA trial assessed the efficacy of the administration of an ARB candesartan, or BB metoprolol in breast cancer patients receiving adjuvant anticancer therapy. It was shown that treatment with candesartan decreased the overall decline in LVEF measured by CMR when compared to placebo, which was not observed in the metoprolol-receiving group. However, the OVERCOME trial, assessing co-treatment with ACEi enalapril and BB carvedilol in patients receiving anthracycline-based chemotherapy for hematologic malignancies, reported mixed results, as a decline in LVED was not prevented in patients undergoing autologous hematopoietic stem cell transplantation. Thus, it remains
somewhat unclear whether neurohormonal blocking agents provide definite cardioprotection against AIC, and more clinical trials are required.”

3. Page 8, line 339 – this section on ARNi in preclinical models of MI can be shortened, since it is of lesser relevance to anthracycline induced cardiotoxicity. This section is actually very nicely summarized in Supplemental Table 1 – consider moving Supplemental Table 1 into the main manuscript, and shortening the text significantly.

Thank you for this suggestion. We have previously considered incorporating Supplemental Table 1 into the manuscript, but have decided not, to because of its large size, causing editorial difficulties with the template format. Due to your comment, we decided to edit out some of the details of experimental animals and gavage protocols, and incorporate the shortened version of the table into the manuscript as Table 1. We also decided to uphold the submission of Supplementary Table 1, however in a shortened version, to provide details missing from the manuscript, for readers that might be interested in them. Additionally, as Table 1 was incorporated, we shortened the text describing the studies on ARNi in MI, editing out the details provided in the Table, and leaving in findings we considered relevant for the purpose of the review (mechanisms of action parallel to mechanisms involved in anthracycline cardiotoxicity).

4. Page 11, line 482 – the clinical relevance of the preclinical studies on HFpEF are unclear, since the PARAGON-HF trial showed no clinical benefit to angiotensin-neprilysin inhibition in HFpEF. Consider shortening the text of this section and moving Supplemental Table 2 into the main text, as in comment #3 above

Thank you for this suggestion. Similarly to the case of Supplemental Table 1, we have previously considered incorporating Supplemental Table 2 into the manuscript, but have decided not to because of its large size, causing editorial difficulties with the template format. Due to your comment, we decided to edit out some of the details of experimental animals and gavage protocols, and incorporate the shortened version of the table into the manuscript as Table 2. We also decided to uphold the submission of Supplementary Table 2, however in a shortened version, to provide details missing from the manuscript, for readers that might be interested in them. Additionally, as Table 2 was incorporated, we shortened the text describing the studies on ARNi in different cardiovascular diseases, editing out the details provided in Table 2, and leaving in findings we considered relevant for the purpose of the review (mainly mechanisms of action parallel to mechanisms involved in anthracycline cardiotoxicity).

5. Page 16, line 735 – in this section on ARNi in human studies on AIC, consider mentioning that hypotension may be a limiting factor in the use of ARNi in the cardio-oncology population. Many cardio-oncology patients have low-normal blood pressures either related to poor oral intake or general deconditioning, and may not tolerate the potent antihypertensive effects of ARNi.

Thank you for this important suggestion. We incorporated this into our manuscript, in the section “Summary and future perspectives”:

“It is important to underline here that one of the potential limitations of ARNi administration in the primary prevention of AIC is resultant hypotension. A significant portion of patients undergoing anti-cancer treatment have low or low-normal blood pressure, which can be associated with their nutritional status, decreased thirst, and poor general health status. As there is considerable risk that those patients might not tolerate ARNi treatment, clinical trials addressing this question are necessary.”

Reviewer 2 Report

The manuscript titled " Neprilysin inhibition in the prevention of anthracycline-induced cardiotoxicity." is a very interesting review in the field of cardioncology. The overall structure is of good quality, methods and paragraphs are acceptable and clear to readers. Authors should improve the manuscript in some parts:

1. a more deep description of neprilysin-mediated cardioprotective effects should be done through the description of NLRP3 and Myd-88 pathways involved ( cite doi: 10.3390/nu10091304.)

2. a description of neprelysin-glucose crosstalk interaction should be made. Authors should describe how antidiabetic drugs like empagliflozin should be useful in cancer patients, associated to sacubitril-valsartan in order to reduce ferroptosis and fibrosis induced by doxorubicin ( cite doi: 10.1186/s12933-021-01346-y. )

3. a more deep description of cytokines involved in these cardioprotective pathway should be performed. 

manuscript will be accepted after minor revision. 

Author Response

Dear Reviewer,

Ref.: cancers-2093839
Neprilysin inhibition in the prevention of anthracycline-induced cardiotoxicity.

We would like to thank you for the careful and thorough reading of this manuscript and for the thoughtful comments and constructive suggestions, which help to improve the quality of this manuscript. We hope that the changes we included will be acceptable and will make our manuscript suitable for publication in Cancers. All the authors have read and approved the final manuscript.

Below we provide our responses to each of the Reviewer’s comments:

1. A more deep description of neprilysin-mediated cardioprotective effects should be done through the description of NLRP3 and Myd-88 pathways involved ( cite doi: 10.3390/nu10091304.)

Thank you for this important comment, which allowed us to significantly expand the scope of your manuscript by more in-depth description of the pro-inflammatory pathways involved in cardiotoxicity of doxorubicin. The subject of NLRP3 and MyD88 pathways was included in several parts of our manuscript:

Molecular mechanisms of AIC:

“Additionally, toll-like receptor 2 (TLR-2) and 4 (TLR-4), members of a family of pattern recognition receptors taking part in innate immunity, with the ability to induce NF-κB signaling by forming complexes with myeloid differentiation primary response 88 (MyD88), have been recently shown to be upregulated in the myocardium of experimental animals, and serum of patients receiving chemotherapy regimens including DOX. Their involvement in AIC pathogenesis was further supported by a recent study on mice model of AIC, were it was shown that TLR-2 knock-out prevented DOX-induced cardiac inflammatory response. The TLR-4/MyD88/NF-κB signaling has been shown to induce production of several pro-inflammatory factors and induce the inflammatory response in the myocardial tissues in several studies. One example is the TLR-4/MyD88/NF-κB induced formation of NLR family pyrin domain containing 3 (NLRP3) inflammasome, which has been previously associated with the anthracycline-induced myocardial inflammatory response.”

ARNi in preclinical models of AIC:

“Ye et al. (2021) have shown that the beneficial effects of ARNi administration in mice model of chronic AIC were parallel to those observed in mice with knock-out of TLR-2 gene. A subsequent in vitro experiment showed that ARNi disrupted the formation of a TLR2-MyD88 complex induced by DOX, inhibiting the TLR-2/MyD88/NF-kB downstream pathway and preventing inflammatory response. In a conference report by Maurea et al. (2022), it was shown that treatment with ARNi alleviated the DOX-induced increase in the cardiac activity of this pathway, with decreased cardiac expression of NLRP3 inflammasome, MyD88, damage-associated molecular patterns (DAMPs) and NF-kB in ANRi-treated mice.”

Additionally, we have cited the article you suggested, in the “AIC prevention” section:

“Other agents, including metformin, statins, and phytochemicals such as resveratrol, allicin, lycopene, curcumin, or polyphenols showed cardioprotective and anti-inflammatory effects in preclinical AIC models,”

2. A description of neprelysin-glucose crosstalk interaction should be made. Authors should describe how antidiabetic drugs like empagliflozin should be useful in cancer patients, associated to sacubitril-valsartan in order to reduce ferroptosis and fibrosis induced by doxorubicin ( cite doi: 10.1186/s12933-021-01346-y. )

Thank you for this comment. We have incorporated this in several parts of the manuscript:

AIC prevention:

“Additionally, recently there has been a growing interest in potential efficacy of sodium-glucose cotransporter 2 (SGLT-2) inhibitors in AIC prevention, due to their antioxidant and anti-inflammatory properties. Quagliariello et al (2021) demonstrated that SGLT-2 inhibitor empagliflozin alleviated the DOX-induced decrease in LVEF and radial and longitudinal strain in non-diabetic mice. This was associated with decreased activity of the aforementioned MyD88/NF-κB/NLRP3 pro-inflammatory pathway and a decrease in myocardial ferroptosis and apoptosis.”

ARNi in preclinical models of AIC:

“Those findings were further supported by a recent conference report, where in an in vitro experiment on human cardiomyocytes treated with DOX or trastuzumab, ARNi was shown to decrease the activity of MyD88/NF-kB/NLRP3 pathway, with synergistic effect of co-treatment with a SGLT-2 inhibitor dapagliflozin.”

3. A more deep description of cytokines involved in these cardioprotective pathway should be performed.

As described above, description of the involved pathways was included in several parts of the manuscript.